# Investigation of Transcriptome Patterns in Endometrial Cancers from Obese and Lean Women

**DOI:** 10.3390/ijms231911471

**Published:** 2022-09-29

**Authors:** Konii Takenaka, Ashton Curry-Hyde, Ellen M. Olzomer, Rhonda Farrell, Frances L. Byrne, Michael Janitz

**Affiliations:** 1School of Biotechnology and Biomolecular Sciences, University of New South Wales, Sydney, NSW 2052, Australia; 2Chris O’Brien Lifehouse, Camperdown, NSW 2050, Australia

**Keywords:** endometrial cancer, RNA-seq, gene expression, molecular signatures

## Abstract

Endometrial cancer is the most common gynaecological malignancy in developed countries. One of the largest risk factors for endometrial cancer is obesity. The aim of this study was to determine whether there are differences in the transcriptome of endometrial cancers from obese vs. lean women. Here we investigate the transcriptome of endometrial cancer between obese and lean postmenopausal women using rRNA-depleted RNA-Seq data from endometrial cancer tissues and matched adjacent non-cancerous endometrial tissues. Differential expression analysis identified 12,484 genes (6370 up-regulated and 6114 down-regulated) in endometrial cancer tissues from obese women, and 6219 genes (3196 up-regulated and 3023 down-regulated) in endometrial cancer tissues from lean women (adjusted *p*-value < 0.1). A gene ontology enrichment analysis revealed that the top 1000 up-regulated genes (by adjusted *p*-value) were enriched for growth and proliferation pathways while the top 1000 down-regulated genes were enriched for cytoskeleton restructure networks in both obese and lean endometrial cancer tissues. In this study, we also show perturbations in the expression of protein coding genes (*HIST1H2BL*, *HIST1H3F*, *HIST1H2BH*, *HIST1H1B*, *TTK*, *PTCHD1*, *ASPN*, *PRELP*, and *CDH13*) and the lncRNA *MBNL1-AS1* in endometrial cancer tissues. Overall, this study has identified gene expression changes that are similar and also unique to endometrial cancers from obese vs. lean women. Furthermore, some of these genes may serve as prognostic biomarkers or, possibly, therapeutic targets for endometrial cancer.

## 1. Introduction

Endometrial cancer (EC), an adenocarcinoma of the uterus, is the fifth most common malignancy in women and the most common gynaecological malignancy in developed countries [1] (Ferlay et al., 2015). The incidence of EC is increasing, particularly in countries undergoing rapid socioeconomic transitions [2]. The main risk factors are the length of exposure to endogenous and exogenous oestrogens, obesity, diabetes, being at an early age at menarche, nulliparity, late-onset menopause, older age, and the use of tamoxifen [3]. EC has been broadly classified into two subtypes (type I and type II) based on histological and epidemiological observations, where patients diagnosed with type II have poorer prognoses than patients diagnosed with type I tumours [4] A more recent classification system based on genomic and proteomic analysis has classified EC into four categories: *POLE* ultramutated, microsatellite instability hypermutated, copy-number low, and copy-number high [5].

The transcriptome is the collection of all RNA transcripts which contain messenger RNAs (mRNAs) and non-coding RNAs including microRNAs (miRNA), long non-coding RNAs (lncRNA), long intergenic non-coding RNAs (lincRNAs), circular RNAs (circRNA), and ribosomal RNA (rRNA). Next-generation sequencing (NGS) has emerged as the primary technology for the unbiased profiling of transcriptomes. RNA sequencing (RNA-Seq) is an NGS tool which uncovers the dynamic nature of the transcriptome, including but not limited to, transcript abundance and the nucleotide sequence of specific RNA transcripts. RNA-Seq has facilitated the rapid sequencing of whole transcriptomes, particularly valuable information contributing to our understanding of diseases such as cancer.

Previous transcriptomic studies have identified some of the molecular underpinnings of EC, providing an insight into its pathogenesis. As such, these findings can be translated into the clinical setting as prognostic and diagnostic biomarkers or through targeted therapies [6,7,8]. For example, one study revealed that the lncRNA taurine upregulated 1 (*TUG1*) was up-regulated in EC compared to its corresponding adjacent normal endometrial tissue [9]. The lncRNA-TUG1 may function as a competing endogenous RNA to regulate *VEGFA* levels by sponging miR-299 and miR-34a-5p. *VEGFA* belongs to a family of vascular endothelial growth factor and is the critical regulator in angiogenesis signalling in a variety of tumours [9]. In another study, the lncRNA colon cancer associated transcript-1 (*CCAT1*) was found to be over-expressed in type I EC, and the knockdown of *CCAT1* decreased cell proliferation and colony formation in HEC-1B EC cells [8]. These findings suggest that lncRNA-CCAT1 may promote tumour formation in EC, supporting findings presented in other studies of various cancers [10,11,12]. Similarly, our previous study demonstrated the lincRNAs *LINC01480*, *LINC00645*, *LINC00891*, and *LINC00702* displayed a highly specific expression for EC compared to normal endometrial tissue while also distinguishing EC from other gynaecological cancers [6]. In a separate study, three signalling pathways: LXR/RXR activation, the neuroprotective role for THOP1 in Alzheimer’s disease, and glutamate receptor signalling were found to shift from being mostly up-regulated to being down-regulated with the increasing cancer stage. As such, these pathways may play a role in cancer progression [13]. This study also demonstrated a substantial down-regulation of genes between early and advanced stage tumours with an altered expression pattern of neuronal signalling pathways and markers [13].

In the current study, we report on transcriptomic alterations in EC tissues compared with matched non-cancerous endometrial tissues from obese and lean postmenopausal women. We explored the distribution of transcript biotypes in EC and non-cancerous endometrial tissues, identified differentially expressed genes, and performed gene ontology enrichment analysis of up- and down-regulated genes (ranked by *p*-values). Overall, our study has identified differentially expressed histone-related genes and up-regulated gene ontology (GO) pathways related to cell growth and down-regulated GO pathways related to cytoskeletal rearrangement in EC tissues compared to adjacent non-tumour endometrial tissues. Hereon matched adjacent non-cancerous endometrial tissue will be referred to as the control tissue.

## 2. Results

A gene was considered expressed if it had a count per million (CPM) value of >0.1. From the cohort of obese women, there were 41,736 genes expressed across 20 EC samples and 41,232 genes expressed in 20 control tissue samples with 38,508 genes common between EC and the control tissue (Figure 1A). Protein coding genes had the most overlapped genes, with only a few uniquely expressed in EC or the control tissue (Figure 1B). The gene category ‘other’ included small non-coding RNAs (snRNA), miscRNAs, and microRNAs (miRNA) as described in the GENCODE Release 31 statistics (see Materials and Methods). Similar observations are made from the lean cohort (Appendix A).

The top 1000 genes from the obese and lean cohort, sorted by fold change, were subjected to principal component analysis (PCA). A separation between EC and the control tissue can be observed (Figure 2). Principal component 1 (PC1) captured 24.65% variance and PC2 captured 10.51% variance between EC and the control tissue. There was no obvious separation between obese and lean cohorts in the PCA.

Prior to differential expression, we applied selection criteria where each gene had to have a CPM value > 0.1 and expressed in ≥25% of the total number of samples in each cohort (obese cohort ≥ 10 samples; lean cohort ≥ 6 samples). There were 34,295 and 33,522 genes which matched these criteria in the obese and lean cohorts, respectively. In the obese cohort, differential expression analysis yielded 12,484 differentially expressed genes having adjusted the *p*-value of <0.1. Of these, 6370 were up-regulated and 6114 were down-regulated in EC. Up- and down-regulated genes are provided in Appendix A, respectively. A volcano plot in Figure 3A shows those differentially expressed genes with respect to EC in the obese cohort. The biotypes of each differentially expressed gene, presented in Figure 3B, indicates that 80.68% of up-regulated genes were protein-coding while 63.26% of down-regulated genes were protein coding. In contrast, only 10.19% of up-regulated genes were lncRNAs and 25.38% of down-regulated genes were lncRNAs. In the lean cohort, there were 6219 differentially expressed genes. Of these, 3196 were up-regulated and 3023 were down-regulated. The complete list of genes that were differentially expressed in the lean cohort are provided in Appendix A. A volcano plot has also been generated for the differentially expressed genes for the lean cohort (Appendix A). Similar to the obese cohort, the majority of up- and down-regulated genes were protein coding (85.7% of up-regulated and 77.87% of down-regulated), while 8.45% of up-regulated genes were lncRNAs and 15.51% of down-regulated genes were lncRNAs (Appendix A).

An independent analysis of the TCGA-UCEC data was performed on 589 samples (35 control and 554 EC) without differentiation between obese and lean. There were 23,706 genes which matched the filtering criterion of CPM > 0.1 in ≥25% of samples (148 samples). Of these, 16,474 genes were differentially expressed with 10,386 up-regulated genes and 6088 down-regulated genes in EC shown in Appendix A, respectively. There were 1958 up-regulated genes and 1679 down-regulated genes in common between TCGA-UCEC and the lean cohort. Between TCGA-UCEC and the obese cohort, there were 3634 up-regulated genes and 2759 down-regulated genes in common.

The top 10 differentially expressed genes in obesity-related EC (top five up-regulated and top five down-regulated) by the adjusted *p*-value were selected for visualisation (Figure 4). All five up-regulated genes are protein coding with four genes encoding histones: H2B clustered histone 13 (*HIST1H2BL*), H3 clustered histone 7 (*HIST1H3F*), H2B clustered histone 9 (*HIST1H2BH*), and H1.5 linker histone, cluster member (*HIST1H1B*). The fifth up-regulated gene selected was TTK protein kinase (*TTK*). Four of the selected down-regulated genes are protein coding genes including patched domain containing 1 (*PTCHD1*), asporin (*ASPN*), proline and arginine rich end leucine rich repeat protein (PRELP), and cadherin 13 (*CDH13*), and the fifth down-regulated transcript is MBNL1 antisense RNA 1 (*MBNL1-AS1*), an lncRNA.

The top 1000 up-regulated and top 1000 down-regulated protein coding genes in EC tissue from obese women were selected by the adjusted *p*-value for a GO enrichment analysis (Figure 5). The enrichment pathways for up-regulated genes showed the enrichment of biological processes related to chromosome segregation, nuclear division, mitotic nuclear division, organelle fission, and nuclear chromosome segregation. Down-regulated genes enrichment categories were related to extracellular matrix organisation, cell-substrate adhesion, the muscle system process, muscle contraction, and extracellular structure organisation. A full list of enrichment gene ontology categories for up-regulated and down-regulated genes, respectively, is shown in Appendix A. Similar GO pathways were enriched in the differentially expressed genes from the lean cohort (Appendix A for up- and down-regulated genes, respectively. Appendix A for up- and down-regulated genes respectively). Further, similar GO pathways were observed in the TCGA datasets in Appendix A for up- and down-regulated genes, respectively.

Following differential expression analysis, common genes (from all biotypes) were identified across obese and lean cohorts. Overall, there were 2727 genes that were up-regulated in both the obese and lean cohorts (*p*-value = 0, hypergeometric test), and 2715 genes that were down-regulated in both the lean and obese cohorts (*p*-value = 0, hypergeometric test), (Figure 6). However, there were more up- and down-regulated genes that were unique to EC tissues from the obese cohort (3642 up-regulated and 3399 down-regulated), compared to the lean cohort (469 up-regulated and 307 down-regulated) (Figure 6). Interestingly, there was one common gene (Clavesin 1; *CLVS1*) that was up-regulated in obese EC and down-regulated in lean EC. However, there were no common genes between obese down-regulated and lean up-regulated (Figure 6). A list of overlapping and unique genes is provided in Appendix A.

## 3. Discussion

In this study, we show the alterations in gene expression in EC from a cohort of 20 obese postmenopausal women and 11 lean postmenopausal women, adding to the increasing body of research into the most common gynaecological cancer. We looked at the biotypes of genes identified across EC and the control tissue, followed by differential expression analysis and a GO enrichment analysis. Furthermore, we identified the common differentially expressed genes from the obese cohort and lean cohort. This study corroborates results from existing studies from our group and the wider scientific community.

In eukaryotes, chromatin is comprised of repeating structural units called the nucleosome core particles which are connected by ‘linker’ DNA. Each nucleosome is composed of 147 base pairs (bp) of DNA wrapped around a core of histone octamers [14]. The histone octamer is made up of a central tetramer of histones H3 and H4, flanked by two dimers of histones H2A and H2B [14]. Interestingly, four of the five top up-regulated genes selected for visualisation were histone encoding genes namely *HIST1H2BL*, *HIST1H3F*, *HIST1H2BH*, and *HIST1H1B*.

Nayak et al. reported that *HIST1H2BL*, along with 21 other histone variants, were significantly over-expressed in recurrent breast tumours (aromatase inhibitor-resistant) compared with the controls [15]. This study showed that the over-expression of histone variants might be important in an endocrine response in oestrogen receptor-positive breast cancer [15]. Another study found *HIST1H3F* to be up-regulated in breast cancer, however, this study had a small sample size to examine the levels of histone members using qPCR (7 clinical BC and 10 adjacent non-cancerous tissues) which was recognised as a limitation [16]. *HIST1H3F* was also identified as one of four genes in a multigene classifier for larynx carcinoma [17] and was proposed to be a novel prognostic biomarker of muscle invasive bladder cancer (MIBC) [18]. Patients whose tissues had high *HIST1H3F* expression levels had significantly longer overall survival than patients who had low *HIST1H3F* expression levels [18]. *HIST1H2BH* along with *PLK1* might serve as a prognostic biomarker of non-small cell lung cancer patients [19]. *HIST1H1B* mRNA expression was significantly higher in breast cancer tissue than normal breast tissue in basal-like breast cancer *(BLBC)* [20]. HIST1H1B protein was also significantly higher in breast cancer tissue compared to normal breast tissue. As such it was proposed that HIST1H1B has the potential to be a therapeutic target of BLBC given its association with breast cancer aggressiveness [20].

The WGCNA R package is a collection of R functions to perform a weighted correlation network analysis (WGCNA) [21]. One function identifies sets of genes, called modules, with similar co-expression patterns. Genes within these modules that are highly connected are termed hub genes and are functionally important [21]. One study identified four hub genes *BUB1B*, *NDC80*, *TPX2*, and *TTK* which were independently associated with the prognosis of EC using a WGCNA algorithm [22]. These four genes were up-regulated in EC, in both obese and lean cohorts, in the current study. Tyrosine threonine kinase (*TTK*) has been shown to be up-regulated in several cancers including lung cancer, gastric cancer, and pancreatic ductal adenocarcinoma, strongly suggesting it may be critical in cell proliferation in tumours [23,24,25]. In another study, *TTK* was significantly up-regulated in an endometrial endometrioid carcinoma, with the knockdown of TTK inhibiting EC cell growth as well as inducing cell apoptosis in EC cell lines (AN3CA and HEC-1-B) [26].

There were 6114 down-regulated genes in EC from the obese cohort in this current study, with the top 10 (by adjusted *p*-value) consisting of 9 protein-coding genes and 1 lncRNA. Of the top 10, patched domain containing 1 (*PTCHD1*) located in the X-chromosome (Xp22.11) encodes the patched domain containing 1 protein and was the most down-regulated by fold-change. The gene is transcribed in the brain, notably in the cerebellum with deletions in this gene associated with autism spectrum disorder and intellectual disability [27]. There is limited information on *PTCHD1* and its involvement in cancer. Katayama et al. identified that *PTCHD1* was expressed in stromal cells from locally advanced breast cancer tumours responsive to chemotherapy [28].

A small leucine-rich proteoglycan (SLRP) is a family of macromolecules found abundantly in the ECM which functions as a structural constituent and as signalling molecule [29,30]. Belonging to this family are asporin (*ASPN*) and proline/arginine rich end leucine rich repeat protein (*PRELP*) which were identified as down-regulated in EC, in both obese and lean cohorts, in this study. Asporin (*ASPN*), encodes a protein found in the extracellular matrix (ECM) of cartilage [31,32]. Zhang et al. found that *ASPN* expression was higher in normal endometrial tissue compared to EC tissue. Additionally, the authors speculate that lncRNA CASC7 inhibits miR-26, inhibits ASPN, inhibits TGF-β, and promotes XIAP, to activate the SMAD/XIAP pathway in EC, thus improving survival and the invasiveness of endometrial cancer cells [33] PRELP protein may function as a molecule anchoring basement membranes to the underlying connective tissue [34].

Another down-regulated gene cadherin-13 (*CDH13*) in obese and lean cohorts, also called H-cadherin or T-cadherin, belongs to the cadherin superfamily and is a cell adhesion molecule and functions as a tumour suppressor gene [35]. The down-regulation of *CDH13* may be related to tumour invasiveness in EC, as it was observed in breast cancer where cells transfected with *CDH13* were less invasive [36]. Muscleblind-like 1 antisense RNA 1 (*MBNL1-AS1*) is an lncRNA that was also down-regulated in EC in obese and lean patients. Its down-regulation has also been observed in prostate cancer and NSCLC whereby the overexpression of *MBNL1-AS1* repressed cell proliferation [37,38]. *MBNL1-AS1* may function as a sponge for miR-181a-5p [37] and miR-135a-5p [38]. lncRNAs have been implicated as competing endogenous RNAs (ceRNAs), whereby these lncRNAs are able to sponge miRNAs to regulate gene expression at the post-transcription level [39].

The Jagged2 (*JAG2*), Aurora Kinase A (*AURKA*), Phosphoglycerate Kinase 1 (*PGK1*), and Hypoxanthine Guanine Phosphoribosyltransferase 1 (*HPRT1*) genes were previously shown to be up-regulated in EC compared to the control samples, with a stepwise elevation in protein expression corresponding to the cancer grade [7] In the present study, all four genes (*JAG2*, *AURKA*, *PGK1*, and *HPRT1*) were significantly up-regulated in EC tissues from obese women. These genes, except *JAG2*, were also up-regulated in EC from lean women. AURKA belongs to the family of serine/threonine kinases and is involved in the regulation of cell cycle progression. In another study, immunohistochemistry showed the overexpression of AURKA in EC tissues compared with control endometrial tissue, and AURKA overexpression was associated with the cancer grade [40]. The knockdown of *AURKA* in HEC-1B cells successfully decreased *AURKA* mRNA and AURKA protein [40]. These findings indicate that *AURKA* mRNA or protein may be suitable biomarkers and a target for EC therapy.

A previous study from our group identified that several long intergenic non-coding RNAs were differentially expressed in EC [6]. *LINC00958*, *LINC01480*, and *LINC00645* were up-regulated in EC whereas *LINC00891* and LINC00702 were down-regulated in the previous study, with the latter four showing a high specificity to EC as compared to the control tissue [6] This study corroborates these observations; *LINC00958* and *LINC00645* were both up-regulated and *LINC00891* and *LINC00702* were both down-regulated in the obese cohort (adjusted *p*-value < 0.1). Whereas in the lean cohort, *LINC00958* was up-regulated and *LINC00891* and *LINC00702* were down-regulated. In another study, the lincRNA metastasis-associated lung adenocarcinoma transcript 1 (*MALAT-1*) was identified in the peripheral blood of patients with non-small cell lung cancer (NSCLC) showing different expression levels between NSCLC patients and the controls [41]. However, despite *MALAT-1* displaying the characteristics of an ideal biomarker (minimally-invasive, exhibiting high specificity, and robustness), *MALAT-1* was unable to discriminate between NSCLS patients and the cancer-free controls with high sensitivity. Thus, the low sensitivity of *MALAT-1* would prevent it from being used as a single biomarker [41] This previous study, however, clearly demonstrates that lincRNAs can be identified in blood, and as such it is possible that lincRNAs specific for EC may be identified and used as biomarkers.

Our GO enrichment analysis of the top 1000 up-regulated genes showed that these genes are mainly involved in cell division/proliferation such as chromosome segregation, nuclear division, mitotic nuclear division, organelle fission, and nuclear chromosome segregation. The top 1000 down-regulated differentially expressed genes were associated with the rearrangement of the cytoskeleton such as extracellular matrix organisation, cell-substrate adhesion, muscle system process, muscle contraction, and extracellular structure organisation. All of these processes were also identified in the lean cohort and are known to play a clear role in cancer development and progression. In contrast, processes related to obesity were not clearly identified in these analyses.

Cervical and ovarian cancer are two other gynaecological malignancies. Several genetic and epigenetic factors associated with these cancers have been identified. For instance, *HOTAIR* has been correlated with cervical cancer recurrence [42,43] as well as the functional role in ovarian, endometrial, and cervical cancers [44]. The copy number and protein expression of claudin-1, CLDN1, was found to increase with the progression of cervical cancer [45]. *PTEN* methylation and loss of *PTEN* expression are early events in the development of cervical cancer [46]. In ovarian cancer, a panel of extracellular vesicle-derived circulating miRNAs may be useful for an early diagnosis [47]. Cancer antigen 125 is a protein encoded by the *MUC16* gene and used in diagnostic tests [48].

In this study, we observed that obese and lean endometrial cancer tissues shared more than 2700 up- and down-regulated genes. This suggests that there is a considerable overlap in the transcriptome of lean and obese EC. However, there were also differences between the two with EC tissues from obese patients expressing over 7000 unique differentially expressed genes (3642 up- and 3399 down-regulated genes) compared with EC tissues from lean patients (469 up- and 307 down-regulated genes). These analyses indicate that many genes that are differentially expressed in EC are related to cancer development, independent of obesity. However, obesity does appear to play a role in the expression of thousands of genes in ECs from obese women. Whether these genes are regulated by obesity-related factors, such as oestrogen, or whether they impact patient prognosis is unclear and warrants further investigation.

The limitations of our study include the small number of samples for both the obese and lean cohorts. Therefore, future studies will investigate whether the differentially expressed genes are also expressed in larger cohorts, and whether these genes influence prognosis in EC. Additionally, in vitro functional studies will be conducted to determine whether altering the expression of the top up- and down-regulated genes identified in this study disrupt EC cell phenotypes such as cell proliferation, cell cycle progression, survival, and migration/invasion.

In conclusion, our study has shown the networks of genes that are dysregulated in EC tissues from obese and lean women. Our findings, along with the growing body of transcriptomic research, may contribute to the identification of diagnostic and prognostic biomarkers or therapeutic targets for this common cancer in women.

## 4. Materials and Methods

### 4.1. Endometrial Cancer and Control Tissue Samples

Women were recruited to our clinical study at the Royal Hospital for Women/Prince of Wales Private Hospital (Randwick, Sydney, Australia) with the inclusion criteria of: being 18 years or older, postmenopausal (ceased having regular periods at least 12 months prior), a BMI of >30 kg/m^2^ (for the obese cohort), and planned hysterectomy for EC (diagnosis of endometrioid adenocarcinoma on curettings, any grade). Consent was received from all patients prior to sample collection, and all processing and experiments were approved by the Human Research Ethics Committee (HREC) of the South Eastern Sydney Local Health District (HREC 15/339). Clinical data were recorded in a de-identified data-base and matching samples were stored at the Lowy Biobank UNSW, Sydney. Sections of benign or malignant endometrial tissue were collected under sterile conditions and immediately placed in 1 mL of Allprotect tissue reagent (Qiagen) in sterile cryovial tubes, and then stored at −80 °C until processed. The endometrial tissues were powdered and the RNA was extracted using an AllPrep DNA/RNA/Protein Mini Kit (Qiagen, Cat # 80004). For the obese cohort, the mean BMI of patients was 38.41 and mean age was 66.7 years. This study originally included 24 obese patients with EC, however, 4 patients were not included as the RNA-seq data were unavailable for the matched pair of tissues [49]. For the lean cohort, the mean BMI of the patients was 24.01 and the mean age was 70.0 years; there were 16 women with EC, however 5 patients were not included as RNA-seq data were unavailable for the matched pair of tissues. The averages provided for age and BMI exclude the samples missing their respective matched RNA-seq data.

### 4.2. RNA Template Preparation and Sequencing

Total RNA was subjected to ribosomal RNA (rRNA) depletion and sequencing library preparation using an Illumina TruSeq Stranded Total RNA Gold kit. Paired-end 126 bp read length sequencing was performed using an Illumina HiSeq 2500 sequencer. A total of 6.3 × 10^9^ paired-end reads were analysed from 40 tissue samples from the obese cohort (20 EC tissue samples and 20 matched adjacent non-cancerous endometrial tissue) and 22 tissue samples from the lean cohort (11 EC tissue samples and 11 matched adjacent non-cancerous endometrial tissue). Sequencing adapters were trimmed by the sequencing facility. FastQC v0.11.8 [50] was performed on the technical replicates and then the concatenated technical replicates. Concatenated technical replicates were utilised in downstream bioinformatic analyses.

The RNA-Seq data files of technical replicates for each biological sample included in this study were concatenated into a single FASTA file per biological sample. As the sequencing adapters were trimmed by the sequencing facility before being received, no trimming was required in this analysis. The samples were aligned to the UCSC hg38 reference genome with HISAT2 and annotated using StringTie. Hereon matched adjacent non-cancerous endometrial tissue will be referred to as the control tissue. Subsequent bioinformatic analysis focused on the differential linear RNA transcriptome in EC and the control tissues including the assignment of a gene biotype based on the gene classification outlined in GENCODE, a principal component analysis (PCA), differential expression analysis, and a GO enrichment analysis.

### 4.3. Sequence Read Alignment and Transcript Assembly

Alignment to the human reference genome (H. Sapiens UCSC hg38) was conducted using an HISAT2 v2.1.0 [51], followed by StringTie [52] to quantify the number of reads mapping to each gene in the reference annotation GENCODE v31. We then used a Python script (prepDE.py) provided with StringTie to generate a read count matrix at the gene level. Genes which had a CPM value of >0.1 were considered expressed. Genes were then classified based on their respective biotype determined by their gene classification as described by Gencode [53]; the classifications were protein-coding, lncRNA, and pseudogene, and the remaining biotypes were classified as ‘other’. Venn diagrams were produced of the biotypes across EC and the control tissue using the Python matplotlib-venn v0.11.7 package [54].

### 4.4. Principal Component Analysis

To visualise the variation between samples, a principal component analysis (PCA) was conducted. Genes were initially filtered with the criteria of a count per million (CPM) value of >0.1 and expressed in ≥1 samples. Subsequently, the top 1000 genes by fold change were selected and a PCA was conducted for dimensionality reduction with PCAtools v2.6.0 [55].

### 4.5. Differential Expression

Differential expression testing was based on normalised read counts, using a trimmed mean of M-values (TMM) with a generalised linear model (GLM) from the Bioconductor package edgeR [56,57]. A pairwise comparison was used as our design matrix describing the comparison between pairs and tissue type (EC and control tissue). A tag-wise dispersion estimate (estimateDisp) was then used by applying glmQLFit to fit a quasi-likelihood negative binomial generalised log-linear model to the normalised read counts of each gene. To test for statistical significance, we used glmQLFTest which uses a quasi-likelihood F-test for coefficients in the linear model. We applied the Benjamini-Hochberg method on the *p*-values to account for multiple testing and to control the false discovery rate (FDR). Genes were determined to be differentially expressed if the adjusted *p*-value was <0.1. The *p*-value is the probability of observing the output data under the null hypothesis.

To visualise differentially expressed genes, a volcano plot was generated with the R package ggplot2 v3.3.2 [58]. Stacked bar charts and CPM scatter and box and whisker plots were generated in GraphPad Prism v8.4.2.

### 4.6. Gene Ontology Enrichment Analysis

We used the top 1000 up-regulated and the top 1000 down-regulated genes (as determined by adjusted *p*-value) to conduct a gene ontology (GO) enrichment analysis utilising clusterProfiler [59] using the enrichGO function.

### 4.7. Venn Diagram for Common Differentially Expressed Genes between Lean and Obese Cohort

The set() data structure in python3 was used to identify the intersection between up- and down-regulated genes across the lean and obese cohort as well as those uniquely expressed to a particular condition (EC or control; up- or down-regulated). Venn diagrams were then drawn manually in Prism with the appropriate values for each section added to the data tables.

### 4.8. Hypergeometric Testing

The R phyper function was used to test the probability of observing overlapping genes by chance from up- or down-regulated genes. A *p*-value < 0.05 was considered significant.

### 4.9. Validation with the Cancer Genome Atlas (TCGA) Data

Gene expression profile data from 589 EC and control samples were downloaded from the TCGA website (https://portal.gdc.cancer.gov/projects/TCGA-UCEC accessed on 7 September 2022). Of the 589 samples, 35 were control samples and the remaining 554 samples were EC. Differential expression was performed with the edgeR ‘limma’ module with a filtration criterial of CPM > 0.1 in ≥25% of the samples (148 samples). Genes with an adjusted *p*-value of <0.1 were considered differentially expressed.

## Figures and Tables

**Figure 1 ijms-23-11471-f001:**
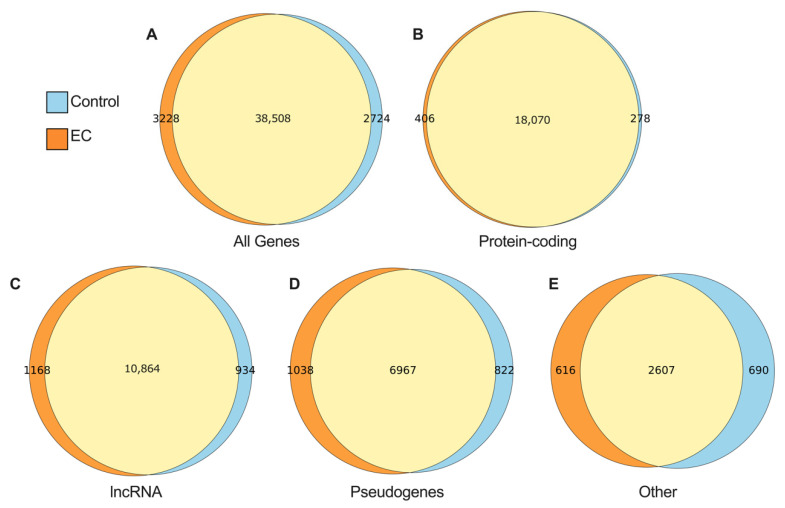
Venn diagrams summarising genes by biotype in EC and control tissue from obese women. Presented are the (**A**) total genes regardless of biotype; (**B**) protein coding genes; (**C**) long non-coding (lncRNA); (**D**) pseudogenes; and (**E**) other. Numbers within Venn diagrams display the number of genes common to both EC and control (overlapping circles in yellow), unique to EC (orange), and unique to control tissue (blue).

**Figure 2 ijms-23-11471-f002:**
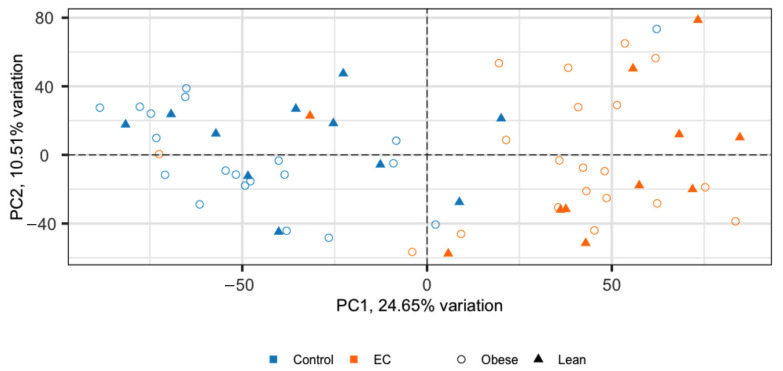
Principal component analysis (PCA) of EC and control tissue of the top 1000 most varied genes by fold-change from obese and lean women. EC samples in orange and control tissue in blue; circles represent the obese cohort and triangles represent the lean cohort.

**Figure 3 ijms-23-11471-f003:**
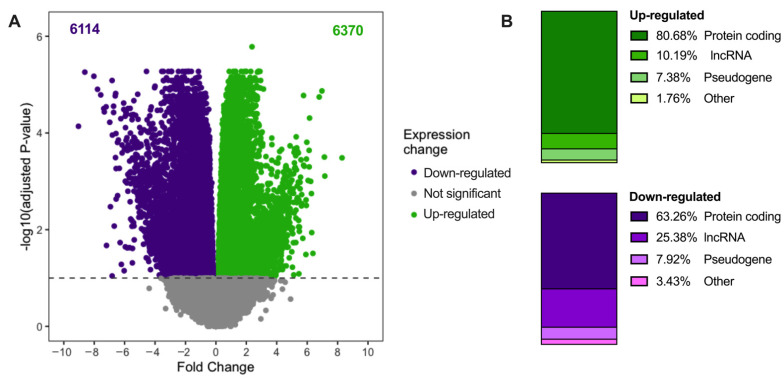
Differential expression analysis between EC and control tissues from the obese cohort. (**A**) Volcano plot of differentially expressed genes. The *x*-axis is the fold-change in gene expression between EC and control tissues and the *y*-axis is the log10 (adjusted *p*-value). Green, purple, and grey dots represent up-regulated, down-regulated, and non-differentially expressed genes in EC, respectively. (**B**) Stacked bar charts show the percentage of each biotype of up-regulated genes in shades of green and down-regulated genes in shades of purple.

**Figure 4 ijms-23-11471-f004:**
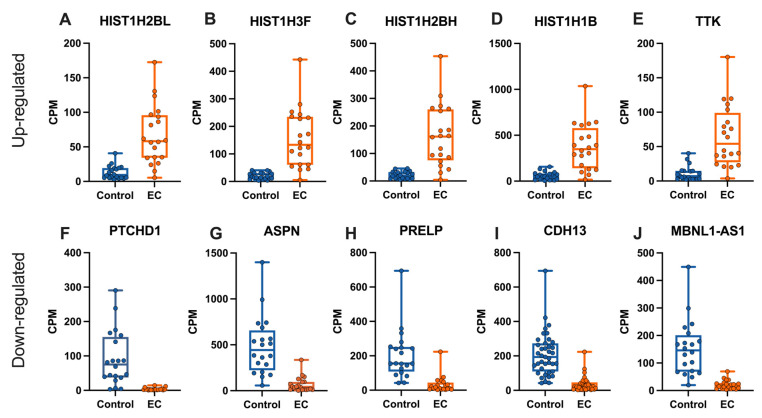
Expression patterns of selected differentially expressed gene between control and EC samples from the obese cohort. Scatter plots and box and whisker plots present expression levels in counts per million (CPM) for EC compared to control tissue with boxes denoting the interquartile range (IQR), median represented by horizontal bar within the IQR, and vertical bars indicating minimum and maximum values. (**A**–**E**) Expression patterns of the top five up-regulated genes in EC and (**F**–**J**) top five down-regulated genes in EC by adjusted *p*-value. The *y*-axis shows gene expression in CPM, dots indicate the individual sample expression values for EC (blue) and control tissue (orange).

**Figure 5 ijms-23-11471-f005:**
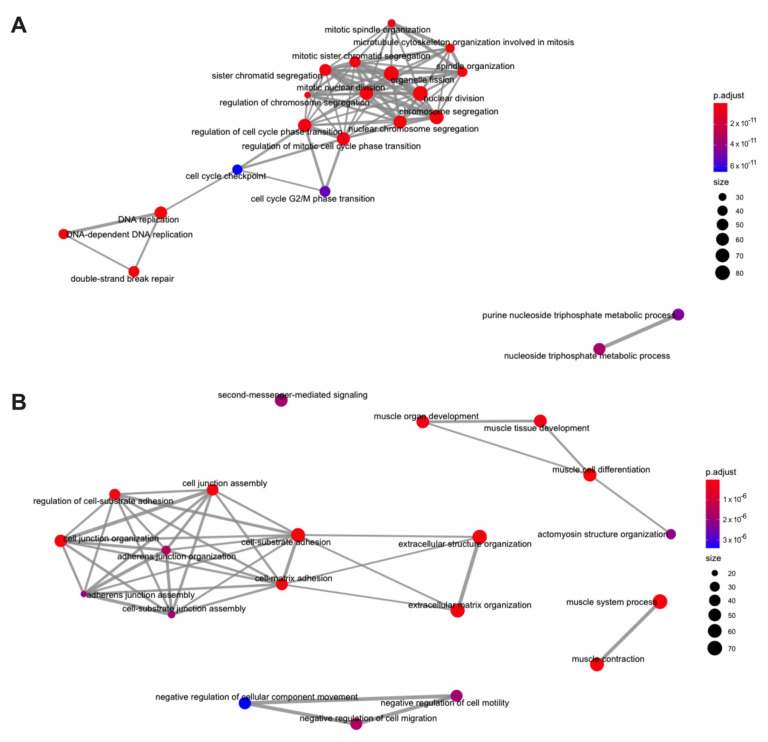
Gene ontology enrichment analysis of protein coding genes in EC from obese women. (**A**) Top 1000 up-regulated genes by adjusted *p*-value. (**B**) Top 1000 down-regulated genes by adjusted *p*-value.

**Figure 6 ijms-23-11471-f006:**
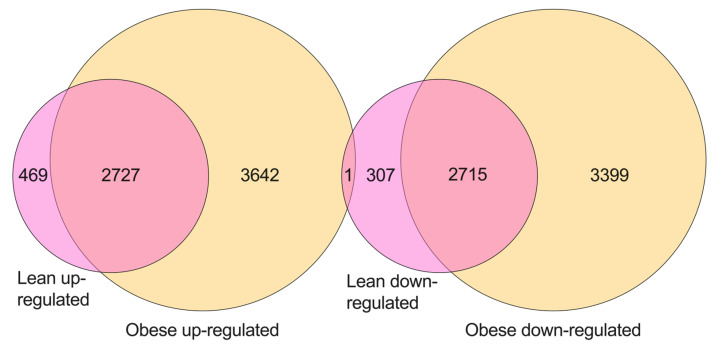
Venn diagram of differentially expressed genes (up- and down-regulated) across obese and lean cohort of women.

## Data Availability

The data presented in this study are available in Appendix A.

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
