# Peer review of "Investigation of Transcriptome Patterns in Endometrial Cancers from Obese and Lean Women"

_ijms, 2022, doi:10.3390/ijms231911471_

Round 1

Reviewer 1 Report

The authors analyzed RNAseq data from the tissues of the EC patients, discovered up / down-regulated genes, and performed gene ontology analysis. They discovered that the top 1000 up-regulated genes were enriched for growth and proliferation while the top 1000 down-regulated genes were enriched for cytoskeleton restructure.  This study could provide insights for EC biologists to follow up on the newly discovered genes. 

It could improve the manuscript if the authors provide a solid discussion about which genes and pathways are promising ones for future follow-up to develop as therapeutic targets based on existing experiments.

Author Response

Reviewer 1

It could improve the manuscript if the authors provide a solid discussion about which genes and pathways are promising ones for future follow-up to develop as therapeutic targets based on existing experiments.

Response

The discussion has now been extended accordingly.

Reviewer 2 Report

I read with great interest this Manuscript, which falls within the aim of the Journal.
In my honest opinion, the topic is interesting enough to attract the readers’ attention. The methodology is accurate, and the data analysis supports conclusions. Nevertheless, authors should clarify some points and improve the discussion by citing relevant and novel critical articles about the topic.

- OVERALL COMMENTS:

- Manuscript should be further revised by a native English speaker to improve clarity and readability.

- I want to inform You that I make a plagiarism check routinely, and I can confirm that Yours is an original writing

- METHODS:

-This section must be improved and placed before the results. As well, some considerations in this part are truly results and must be moved. I suggest rewriting this part and making it possible for other researchers to repeat the study.

- DISCUSSION:
- The authors have not adequately highlighted the strengths and limitations of their study. I suggest better specifying these points
- What are the actual clinical implications of this study? it is essential to report the results obtained by the authors in the context of clinical practice and to adequately highlight what contribution this study adds to the literature already existing on the topic and to future study perspectives

Author Response

Reviewer 2

- Manuscript should be further revised by a native English speaker to improve clarity and readability.

Response

The manuscript has been edited by a native English speaker.

- This section must be improved and placed before the results. As well, some considerations in this part are truly results and must be moved. I suggest rewriting this part and making it possible for other researchers to repeat the study.

Response

According to the journal’s manuscript template the Materials and Methods section follows Discussion. We also believe that Materials and Methods section, in its present forms, allows to repeat the study by independent researchers.

- The authors have not adequately highlighted the strengths and limitations of their study. I suggest better specifying these points

Response

The limitations of the study have now been addressed, accordingly. This has been addressed in the discussion, 2nd last paragraph.

- What are the actual clinical implications of this study? it is essential to report the results obtained by the authors in the context of clinical practice and to adequately highlight what contribution this study adds to the literature already existing on the topic and to future study perspectives

Response

We acknowledge the reviewer’s comment as highly relevant and important. We however feel that the gene targets, specific for EC and presented in our study are at too early stage of evaluation to be considered for clinical relevance of this malignancy. We believe that our study paves the way for future investigations in more clinically oriented set-up.

Reviewer 3 Report

The authors presented transcriptome patterns in obesity related endometrial cancer. The manuscript is well written and presented important data particularly  the differential expression difference between adjacent normal tissue and endometrioid endometrial cancer (EEC). However, there are some concerns that needs to be addressed.

1. The sample size is very small and needs justification for limiting the analyses for only 20  samples of EEC. Did the authors calculate the power during experimental design?

2. Why the authors limited only the analyses to obese women with endometrial cancer. It would be more informative if the analyses were performed in both obese and none obese women to understand the differences in pathological pathways. Such analyses could give if obesity specific gene signatures are implicated in the pathogenesis of EEC and these data could be helpful for prevention or management of EEC.

4. The authors need to report the clinicopathologic pathern of the 20 cases and assess if the transcriptome differ by grade and stage of EEC  particularly in JAG2, AURKA, PGK1, HPRT1, LINC00958, LINC00645, LINC00891 and LINC00702.

5. This finding needs validation cohort. I suggest to look the publicly available UCEC TCGA Genome data to validate these findings.

6. It would be interesting to investigate the role of some of the up-regulated or down regulated genes (e.g JAG2, AURKA,  PGK1, HPRT1, LINC00958, LINC00645, LINC00891 and LINC00702) in EEC cell lines if implicate in growth and proliferation using in vitro experiment.

Author Response

Reviewer 3

  1. The sample size is very small and needs justification for limiting the analyses for only 20 samples of EEC. Did the authors calculate the power during experimental design?

Response

According to Hard et al. (J Comput Biol. 2013 Dec; 20(12): 970–978) sample size of n=20 in each condition allows for detection of differentially expressed genes with fold change of 3 or over, which in our study comprises all discussed top differentially expressed genes. Accordingly, we believe that our sample size was sufficient to make our observations representative.

  1. Why the authors limited only the analyses to obese women with endometrial cancer. It would be more informative if the analyses were performed in both obese and none obese women to understand the differences in pathological pathways. Such analyses could give if obesity specific gene signatures are implicated in the pathogenesis of EEC and these data could be helpful for prevention or management of EEC.

Response

We agree with the reviewer that extension of the investigation onto tissues samples derived from lean women would enrich conclusiveness of the study. Accordingly, we now included an analysis of lean women-derived samples presented in Results section of the revised manuscript.

  1. The authors need to report the clinicopathologic pattern of the 20 cases and assess if the transcriptome differ by grade and stage of EEC particularly in JAG2, AURKA, PGK1, HPRT1,LINC00958, LINC00645, LINC00891 and LINC00702.

Response

Accordingly, we included the EC grade information for each sample in supplementary material for up- and down-regulated genes for lean and obese groups.

  1. This finding needs validation cohort. I suggest to look the publicly available UCEC TCGA Genome data to validate these findings.

Response

We agree with the reviewer that such validation should be performed, and this will stay in focus of our future investigations.

  1. It would be interesting to investigate the role of some of the up-regulated or down regulated genes (e.g JAG2, AURKA, PGK1, HPRT1, LINC00958, LINC00645, LINC00891 andLINC00702) in EEC cell lines if implicate in growth and proliferation using in vitro experiment.

Response

Again, we agree with the point that the study would benefit from functional validation of individual targets. Unfortunately, we currently don’t have enough resources and sample material to perform this type of experimentation.

Round 2

Reviewer 2 Report

  1. I read with great interest this Manuscript, which falls within the aim of the Journal.
    Honestly, the topic is interesting enough to attract the readers’ attention. The methodology is accurate, and the data analysis supports conclusions. Nevertheless, authors should clarify some points and improve the discussion by citing relevant and novel critical articles about the topic.

    - OVERALL COMMENTS:

    - Manuscript should be further revised by a native English speaker to improve clarity and readability.

    - I want to inform You that I make a plagiarism check routinely, and I can confirm that Yours is an original writing

    - METHODS:

    -It is mandatory to move the methodology section just after the introduction to allow readers to understand how samples were collected and from whom.

    The results section is not conceived to contain this kind of information. Please correct it. As well, inclusion and exclusion criteria must be emphasized.

    As well, a statistic analysis section must be considered. Statistic value must be reported when comparing results

    - DISCUSSION:
    - The authors have not adequately highlighted the strengths and limitations of their study. I suggest better specifying these points
    - What are the actual clinical implications of this study? it is essential to report the results obtained by the authors in the context of clinical practice and to adequately highlight what contribution this study adds to the literature already existing on the topic and to future study perspectives
    - To improve comparison with existing literature, authors should better describe the role of a well-known prognostic subset of gene expression for other gynecological oncological diseases (see PMID: 33279812; 33228245)

Author Response

Reviewer 2

Manuscript should be further revised by a native English speaker to improve clarity and readability.

Response

The manuscript has been edited by a native English speaker.

It is mandatory to move the methodology section just after the introduction to allow readers to understand how samples were collected and from whom. The results section is not conceived to contain this kind of information. Please correct it. As well, inclusion and exclusion criteria must be emphasized. As well, a statistic analysis section must be considered. Statistic value must be reported when comparing results

Response

Following reviewer’s suggestion, we expanded description of our bioinformatics pipeline as well as added sections describing statistical calculations of differential expression and hypergeometric test. Regarding location of Materials and methods within the manuscript, according to the journal’s manuscript template, this section follows Discussion.

The authors have not adequately highlighted the strengths and limitations of their study. I suggest better specifying these points

Response

As per our previous response, the limitations of the study have been addressed, accordingly. This has been addressed in the discussion, 2nd last paragraph.

What are the actual clinical implications of this study? it is essential to report the results obtained by the authors in the context of clinical practice and to adequately highlight what contribution this study adds to the literature already existing on the topic and to future study perspectives

Response

As per our previous response, we acknowledge the reviewer’s comment as highly relevant and important. We however feel that the gene targets, specific for EC and presented in our study are at too early stage of evaluation to be considered for clinical relevance of this malignancy. We believe that our study paves the way for future investigations in more clinically oriented set-up.

To improve comparison with existing literature, authors should better describe the role of a well-known prognostic subset of gene expression for other gynecological oncological diseases (see PMID: 33279812; 33228245).

Response

The suggested references have now been cited and discussed in the Discussion section.

Reviewer 3 Report

Thank you for submitting the revised version of the manuscript. I believe the revised version is notably improved in content and presentation. Although the majority of the questions I raised during the first review are replied, Q5 validation of the study deploying  the publicly available UCEC TCGA Genome data is not reasonably addressed by the authors. Generating data from TCGA is the minimum effort that the authors can do to enrich and validate the presented data. I strongly recommend to do the validation using the TCGA genome data. There are are several important genomic and clinical data that can be extracted from the TCGA data. Please report 1) if a similar findings would be generated on differentially  expressed genes and gene ontology in obese and non-obese women with endometrial cancer. In  TCGA data, there are samples taken from adjacent normal endometrial tissue and this could be taken as a control. To have more robust finding the top significantly up-regulated and down-regulated reported genes should be correlate with clinicopathologic parameters, TCGA molecular subtypes and survival outcome.

Minor comment

In the references the author list is too long, for example, Levine, D.A et al. I suggest to limit the number of authors to 6 as most style of reference including the Vancouver method.

Author Response

Reviewer 3

I strongly recommend to do the validation using the TCGA genome data. There are are several important genomic and clinical data that can be extracted from the TCGA data. Please report 1) if a similar findings would be generated on differentially  expressed genes and gene ontology in obese and non-obese women with endometrial cancer. In  TCGA data, there are samples taken from adjacent normal endometrial tissue and this could be taken as a control. To have more robust finding the top significantly up-regulated and down-regulated reported genes should be correlate with clinicopathologic parameters, TCGA molecular subtypes and survival outcome.

Response

Following the reviewer’s comment, we performed differential expression and GO terms enrichment analysis of 589 samples retrieved for the TCGA repository with outcomes presented in the Results section. Furthermore, we performed correlation analysis between up- and down-regulated genes in samples derived from obese and lean individuals, respectively. We also included description of applied methodology for this additional analysis in Materials and methods section.

In the references the author list is too long, for example, Levine, D.A et al. I suggest to limit the number of authors to 6 as most style of reference including the Vancouver method.

Response

Accordingly, we shortened the author lists in all references.

Round 3

Reviewer 2 Report

Thank You for having changed the manuscript according to my consideration. In the present form, it is fine and interesting

Reviewer 3 Report

Thank you for submitting the revised version of the manuscript. The revised manuscript has improved with the addition of TCGA data. The finding in this study seems  different from the previous Proteomic profiling by Mauland et al.

It would be interesting  if the authors include this in the discussion.